



# A two-year intercomparison of CW focusing wind lidar and tall mast wind measurements at Cabauw

Steven Knoop[1], Fred C. Bosveld[1], Marijn J. de Haij[1], and Arnoud Apituley[1]

[1]Royal Netherlands Meteorological Institute (KNMI), Utrechtseweg 297, 3731 GA De Bilt, The Netherlands

**Correspondence:** Steven Knoop (steven.knoop@knmi.nl)

**Abstract.** A two-year measurement campaign of the ZephIR 300 vertical profiling continuous-wave (CW) focusing wind lidar has been carried out by the Royal Netherlands Meteorological Institute (KNMI) at the Cabauw site. We focus on the (height-dependent) data availability of the wind lidar under various meteorological conditions and the data quality through a comparison with in situ wind measurements at several levels in the 213-m tall meteorological mast. We find an overall
availability of quality controlled wind lidar data of 97 % to 98 %, where the missing part is mainly due to precipitation events exceeding 1 mm/h or fog or low clouds below 100 m. The mean bias in the horizontal wind speed is within 0.1 m/s with a high correlation between the mast and wind lidar measurements, although under some specific conditions (very high wind speed, fog or low clouds) larger deviations are observed. The mean bias in the wind direction is within 2°, which is on the same order as the combined uncertainty in the alignment of the wind lidars and the mast wind vanes. The well-known 180° error in
the wind direction output for this type of instrument occurs about 9 % of the time. A correction scheme based on data of an auxiliary wind vane at a height of 10 m is applied, leading to a reduction of the 180° error below 2 %. This scheme can be applied in real-time applications in case a nearby, freely exposed, mast with wind direction measurements at a single height is available.

# 1 Introduction

Atmospheric motion and turbulence are essential parameters for weather and topics related to air quality. Therefore, wind profile measurements play an important role in atmospheric research and meteorology. One source of ground-based wind profile data are Doppler wind lidars, which are active, laser-based remote sensing instruments that measure wind speed and wind direction up to a few hundred meters or even a few kilometers. Like traditional radar wind profilers, Doppler wind lidars
typically cover the atmospheric boundary layer very well and thereby complement other sources of wind information, such as in-situ measurements at surface stations, weather radars, aircraft observations and satellite instruments.

   Doppler wind lidars measure the Doppler-shift of the backscattered laser light by molecules or aerosols in the moving air, by means of either direct detection or coherent detection. This Doppler-shift ($\delta f$) provides the wind velocity along the line-




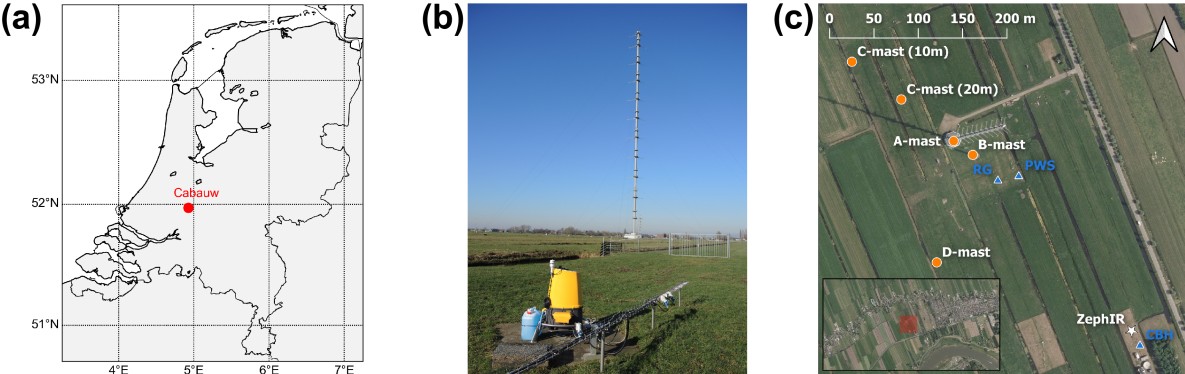

**Figure 1.** (a) Map of the Netherlands, indicating the location of measurement site near Cabauw ($51.971°$ N, $4.927°$ E). (b) Photo of the ZephIR 300 wind lidar instrument, with the 213-m tall A-mast visible the background (view in NW direction). (c) Overview of the locations of the masts, the ZephIR wind lidar and other relevant instruments: rain gauge (RG), ceilometer (CBH) and present-weather sensor (PWS); the inset shows the area around the site (images from PDOK Landelijke Voorziening Beeldmateriaal, Luchtfoto 2019 Ortho 25cm RGB).

of-sight ($V_{\mathrm{LOS}}$) via $\delta f = -2V_{\mathrm{LOS}}/\lambda$, where $\lambda$ is the laser wavelength. Direct detection wind lidars measure the shift of the

frequency spectrum of the return signal. As an example, the ADM-Aeolus space-based wind lidar, launched in 2018, is a direct detection system at a laser wavelength of 355 nm (Stoffelen et al., 2005). In coherent detection the return signal is optically mixed with the local oscillator laser and the resulting beat-signal provides the Doppler-shift. In heterodyne coherent detection the local oscillator is frequency shifted from the transmitted light and the beat frequency has a fixed offset. This frequency shift is absent in homodyne coherent detection and the beat frequency is the absolute value of the Doppler-shift. An extensive and

comprehensive description of the history and fundamentals of wind lidars can be found in Henderson et al. (2005).

Commercial coherent detection wind lidars based on the telecom wavelength of 1.5 $\mu$m became available early 2000's. These systems rely solely on the aerosol signal and their range is typically limited to the atmospheric boundary layer. They are nowadays extensively used within the wind energy industry, for instance for wind resource assessment and wind turbine power curve validation (Mikkelsen, 2015). For national meteorological services, like the Royal Netherlands Meteorological

Institute (KNMI), data sets measured by these instruments can be valuable for model validation, while real-time access opens the possibility of data assimilation in operational numerical weather prediction (NWP) models and nowcasting purposes. For these applications it is of utmost importance to know the meteorological conditions in which the instruments are able to provide reliable data or not.

Here we present results of a two-year measurement campaign of the ZephIR 300 short-range vertical profiling wind lidar at

Cabauw. We focus on the (height-dependent) data availability of the wind lidar under various meteorological conditions and the data quality through a comparison with in situ wind measurements at several levels in the 213-m tall meteorological mast. This wind lidar instrument, and its predecessors, have been extensively tested (see e. g. Smith et al. (2006); Kindler et al. (2007); Peña et al. (2009); Wouters and Wagenaar (2016)). Our campaign is unique in terms of (1) duration (more than two years), (2)



**Table 1.** Overview of the deployed ZephIR 300 wind lidar instrument.

| | |
|---|---|
| laser wavelength | 1.54 $\mu$m |
| ranging | CW focusing |
| horizontal wind retrieval | VAD scan |
| firmware | 2.1027 ZP300 |
| measuring heights | 11 m, 20 m, 39 m, 80 m, 140 m, 200 m, 252 m *above ground level* |
| | (from 2019-05-29 also 60 m, 100 m, 180 m and 300 m) |
| scan dwell time | 1 s |
| height instrument | 1 m |
| meteo station | AIRMAR WeatherStation 200WX |

ability to cover the full height range of the wind lidar due to the 213 m tall mast, and (3) ability to relate the performance of the

instrument to the meteorological conditions due to the co-location of many meteorological instruments on site. The location of the measurement campaign is shown in Fig. 1(a), a photo of the wind lidar with the tall mast in Fig. 1(b), and an overview of the Cabauw site in Fig. 1(c).

This paper is organized as follows. We introduce the wind lidar instrument in Sect. 2, the Cabauw site in Sect. 3 and the measurement campaign in Sect. 4. Results of the intercomparison study are given in Sect. 5, focused on the 10-minute averaged

horizontal wind speed and wind direction data. Finally, we conclude and give an outlook in Sect. 6.

## 2   Wind lidar instrument

The vertical profiling wind lidar of the measurement campaign is the ZephIR 300 (ZX Lidars, UK, formerly ZephIR Lidar). The ZephIR 300 is a homodyne coherent detection CW focusing wind lidar. The laser beam is transmitted through a constantly rotating prism (wedge) to perform a so-called velocity azimuth display (VAD) scan. The scanning cone angle is 30° (with

respect to zenith). For each height one complete rotation takes 1 s, in which 50 measurements of 20 ms are taken, from which the 3D wind vector is reconstructed (i.e. horizontal and vertical wind speed, and wind direction). The manufacturer specifies the wind speed and wind direction accuracies as better than 0.1 m/s and 0.5°, respectively, and a wind speed range from <1 m/s to 80 m/s. The height range is 10-200 m above the instrument, although up to 300 m can be selected in the software. There is a maximum of 10 user-configurable measuring heights, besides a pre-fixed height of 38 m above the instrument, which all are

measured sequentially by changing the focus of the laser beam after each VAD scan. An overview of properties and settings of the deployed ZephIR 300 instrument is given in Table 1. The ZephIR 300 has an automatic wiper system that operates when it rains, and which is also supplied with a washer pump to aid cleaning in case the top window is soiled.

The wind retrieval from the VAD scan is based on the assumption of a homogeneous wind field in the scanning cone at each measurement level. Being a CW focusing wind lidar, the probe length increases quadratically with height: at 10 m height above





the instrument the probe length is 0.07 m, whereas at 200 m it is 30 m. CW focusing wind lidars can be sensitive to clouds that are above the maximum range, as the contribution to the Doppler signal from clouds in the tail of the laser pulse profile can be comparable to the aerosol signal at the preselected focusing height (Smith et al., 2006). A cloud removal algorithm is used to correct for this effect, which involved a measurement at an additional higher altitude (Kindler et al., 2007; Courtney et al., 2008).

As a result of the homodyne detection, meaning that only the absolute value of the Doppler-shift is measured and not the sign, there is a 180° ambiguity in the measured wind direction, as well as a sign ambiguity in the vertical wind speed. To overcome this issue the ZephIR 300 includes a meteo station containing a sonic anemometer, which wind direction information is directly fed into the ZephIR 300 internal algorithm to decide on the true wind direction. This aspect will be further discussed below in Sect. 5.2.3. In addition, zero wind speed cannot be measured, as a small region around zero Doppler shift needs to be filtered

out (Courtney et al., 2008). A more extensive introduction of the ZephIR 300 (and its predecessors) is given in Pitter et al. (2015).

The instrument reports, besides unaveraged data, quality controlled (QC) 10-minute averaged data, including horizontal and vertical wind speed, and wind direction, minimum, maximum and standard deviation of the horizontal wind speed, and turbulence intensity. For wind speed the mean is taken to derive the 10-minute averaged data, for wind direction vector averaging is

applied. Reasons for not passing QC can be a very low wind speed event (<1 m/s), partial obscuration of the window, or significant interference with the laser beam at the specified height, or atmospheric conditions which adversely affect lidar wind-speed measurements. The analysis in this paper is based on this QC 10-minutes averaged data and is focused on the horizontal wind speed and wind direction.

Note that here the marine version of the ZephIR 300, ZephIR 300M, is used, because this measurement campaign is related

to offshore deployment. However, there is no difference in functionality or performance between these versions. Throughout this paper we will address the instrument as ZephIR 300.

## 3   Measurement site

The Cabauw Experimental Site for Atmospheric Research (CESAR) is located in an extended and flat polder landscape, 0.7 m below mean sea level (51.971° N, 4.927° E; see Fig.1(a)). The site is centered around the 213 m research tower ("A-mast")

as shown in Fig.1(b), from which the atmospheric boundary layer can be sampled at various altitudes. Close to the tower is a platform with ground based remote sensing instruments for vertical profiling and column integrated observations, a platform for radiation measurements and a platform for monitoring of the energy balance (land-atmosphere interaction). Together, the instruments in the tower and the surrounding platforms constitute a comprehensive suite for atmospheric monitoring and process studies. The Cabauw site is a National Facility of ACTRIS (Aerosol, Clouds and Trace Gases Infra-Structure) and

ICOS (Integrated Carbon Observation System) and is the main site for the Ruisdael Observatory. An overview of 50-year Cabauw observations and research is given in Bosveld et al. (2020).





**Table 2.** Overview of other meteo instruments used in this study (RSS: remote sensing site, AWS: automatic weather station), their locations are also indicated in Fig. 1(c).

| measurement | location | instrument | height | distance to wind lidar |
|---|---|---|---|---|
| wind speed/direction | A-mast | KNMI cup anemometers/wind vanes | 40 m, 80 m, 140 m, 200 m | 293 m |
| | B-mast | | 10 m, 20 m | 267 m |
| | C-mast | | 10 m, 20 m | 437 m, 367 m |
| | D-mast | | 10 m | 233 m |
| visibility | A-mast | Biral SWS100 | 40 m, 80 m, 140 m, 200 m | 293 m |
| | B-mast | | 2 m, 10 m, 20 m | 267 m |
| precipitation intensity | AWS | KNMI rain gauge | | 226 m |
| cloud base height | RSS | Lufft CHM15K ceilometer | | 19 m |
| present-weather and visibility | AWS | Vaisala FD12P | 2 m | 215 m |

Wind speed and wind direction are measured with KNMI cup-anemometers and KNMI wind vanes, respectively, at six levels: 10 m, 20 m, 40 m, 80 m, 140 m and 200 m, using different masts (see Fig. 1(c) and Table 2). Precautions are taken to avoid too large flow obstruction from the A-mast and the main building at the bottom of the A-mast. At the levels 40 m,

80 m, 140 m, and 200 m of the A-mast the wind direction is measured at three booms and wind speed is measured at two booms. Depending on wind direction, the best exposed sensors are chosen. At the levels 10 m and 20 m the wind direction and wind speed are measured at separate, smaller masts south ("B-mast", SE from A-mast) and north (two "C-masts", NE from A-mast for 20 m and 10 m level, respectively) of the main building; the selection between these two masts depends on the wind direction. The wind data is quality controlled, including corrections for remaining flow distortions from the mast. In addition,

another 10 m mast ("D-mast", South of A-mast) is present. For the C- and D-masts the cup and vane are attached on top of masts, such that flow corrections are not needed.

The KNMI cup anemometer contains a photo-chopper with 32 slits. The sensitivity is 1.98 m of air per rotation, which results in 62 mm air passed per pulse. The distance constant is 2.9±0.4 m. The cup-anemometer measures the length of the wind vector. The accuracy of the cup anemometer is 1 %, or 0.1 m/s for low wind speeds. The cup anemometers are calibrated over a wind

speed range of 2 m/s to 20 m/s. At higher wind speeds the calibration of the cup anemometer maybe slightly non-linear, the main reason being deformation of the cups. However, this effect will only give gradual deviations with increasing wind speed above 20 m/s (see Fig. 12 of Wauben (2007)). During operation, the comparability of the cup anemometers is monitored to stay within 1 % by comparing the two available instruments at the same height, provided that wind direction allows for proper wind measurements for both. Calibration period for the cup anemometer is 14 months. The 10-minute averaged wind speeds

are calculated from the pulses counted in the 10-minute period.

The cup anemometer is calibrated in the laminar flow of a wind tunnel. In a turbulent flow, as is often encountered in the atmosphere, overspeeding occurs because the response time of the instrument is proportional to the wind speed. Vertical





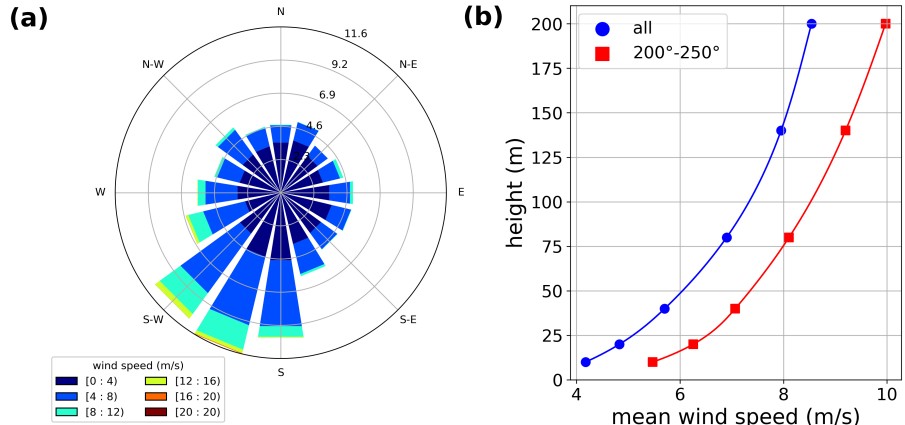

**Figure 2.** Wind conditions during the measurement campaign (Feb.˜15, 2018 until Feb.˜29, 2020): (a) wind rose at 10 m height, (b) wind speed profiles, for all wind directions and the wind sector between 200° and 250°. All based on mast wind measurements.

fluctuations in the flow will also lead to overspeeding as the open sides of the cups will be better exposed to the wind. Following Kristensen (1993) we estimate overspeeding for the KNMI cup anemometer and for neutral conditions as: 1.5 % at 10 m, 0.9 % at 20 m, 0.6 % at 40 m, 0.4 % at 80 m, 0.3 % at 140 m, and 0.3 % at 200 m. These values will be larger under unstable stratification and lower under stable stratification. No corrections are performed for overspeeding.

The KNMI wind vane contains a 8 bits code disk which results in a resolution of 1.5°. The damping ratio is 0.30 and the damped wave length is 7.0 m. Accuracy of the vane depends on the instrument and on orientation of the vane plug. The vane fulfills the WMO requirement of 3°. An overall check of the three vanes at each measurement level in the A-mast suggests a comparability of 2°. Calibration period for the wind vane is 26 months. The wind direction is sampled at 4 Hz and decomposed as x- and y-components of the unit vector. The 10-minute averaged wind direction is derived from the 10-minutes averages of the x- and y-components.

A KNMI automatic weather station (AWS) is located 100 m SE of the A-mast. The AWS includes, among other instruments, a KNMI electrical rain gauge that measures precipitation intensity, and a Vaisala FD12P present-weather sensor that provides measurements of precipitation type and visibility (at a height of 2 m above ground level). The D-mast is part of the AWS. Also part of the AWS, but located at the remote sensing site (RSS), is a Lufft CHM15K ceilometer, which reports the cloud base height (of maximally three cloud layers). In addition, Biral SWS-100 visibility sensors are located in the A-mast (at 40 m, 80 m, 140 m and 200 m) and in the B-mast (at 2 m, 10 m and 20 m). An overview of all relevant instruments, and their distances from the wind lidar, is given in Table 2. An updated description of the in-situ observation program in Cabauw is provided by Bosveld (2020).





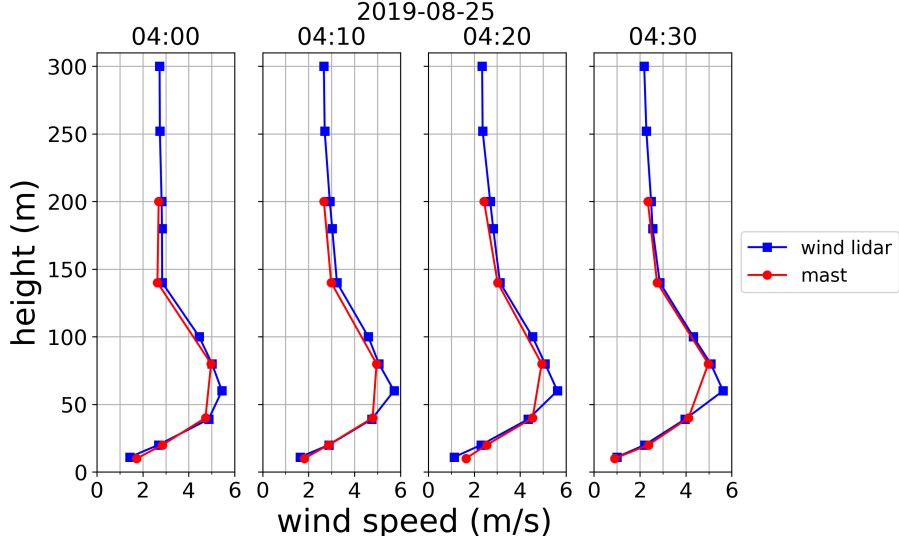

**Figure 3.** Example of a low-level jet in the early morning of August 25, 2019, highlighting the benefit of more measuring levels provided by the wind lidar compared to the mast. The times indicated on top of the panels are the start of the 10-minute interval.

## 4 Measurement campaign

The ZephIR 300 is placed at the northern part of the remote sensing site (RSS) of the Cabauw site, 290 m in SE direction from the A-mast. The estimated accuracy of the alignment of the wind lidar instrument is about 1°. The instrument is configured to measure at (or close to) vertical levels of the mast wind measurements, see Table 1. Note that the minimum range of the
ZephIR 300 is 10 m above the instrument, corresponding to a minimum height of 11 m. Also, the pre-fixed height of 39 m (above ground level) does not allow to select 40 m (as heights should be apart by at least 5 m). The mast wind measurements are interpolated to match those wind lidar measurement levels (see Sect. 5.2). In Fig. 2 the wind conditions during the measurement campaign are shown: (a) the windrose at a height of 10 m and (b) wind speed profile. The ZephIR 300 was operational from February 2018 until June 2020. The data considered here are from February 15, 2018 until February 29, 2020, covering more
than two years. During the measurement campaign two adjustments were made to the wind lidar. On August 22, 2018 the meteo station of the wind lidar was relocated to a separate pole (see Sect. 5.2.3) and on May 29, 2019, the number of measuring levels of the wind lidar was extended from 7 to 11. No maintenance (other than the automatic wiper system) was applied to the wind lidar during the full duration of the measurement campaign.

The maximum amount of measuring heights available in the wind lidar exceeds that of the mast. As a result, the wind lidar
can resolve the wind profile better than the mast. This is in particular relevant for non-monotonic wind profiles, of which low-level jets (LLJ) are the most prominent ones. In Fig. 3 an example of a LLJ is shown, following the criteria by Baas et al. (2009), where the wind lidar clearly captures the maximum wind speed of the LLJ much better than the mast measurements.





An example of a single day comparison between the wind lidar and the mast is shown in Fig. 4. During this day of Feb. 9, 2020, when extratropical cyclone Ciara[1] passed Cabauw, the mast measurements reported the highest reported wind speeds
for all levels during the measurements campaign. The wind lidar and mast measurements are in close agreement for most of the levels, with the exception of some part of the day where the wind speed was around 25 m/s or higher, occurring mostly at 140 m and 200 m. The intercomparison at high wind speeds will be further discussed in Sect. 5.2.1.

---

[1] also named Sabine in Germany or Elsa in the Scandinavian countries



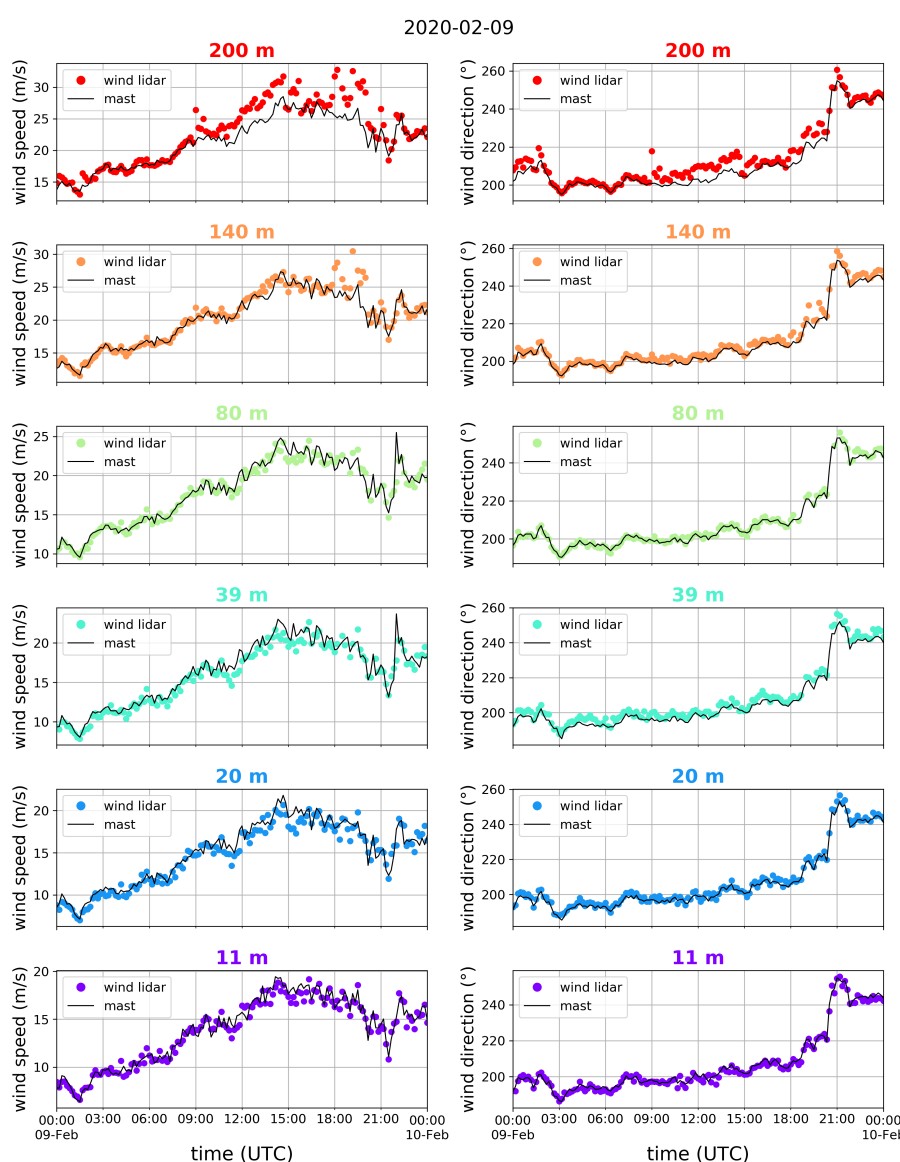

**Figure 4.** Wind measurements of February 9, 2020 (Storm Ciara), comparing the 10-min averaged data of the mast (black solid lines) and the wind lidar (colored symbols), showing wind speed (left) and wind direction (right) for different heights (indicated on top of each panel).





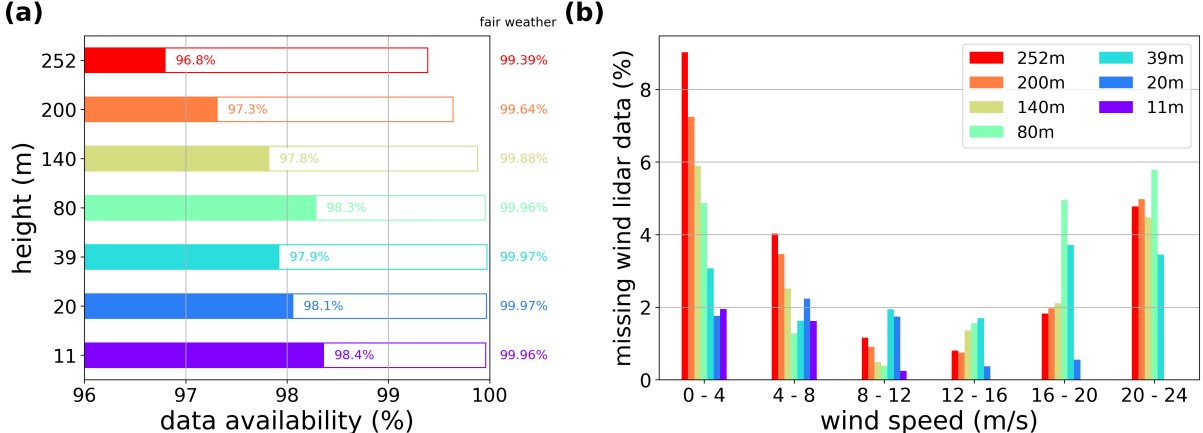

**Figure 5.** (a) Overall availability of QC wind lidar data for different vertical levels, during the full uptime of the instrument (closed bars) and for "fair weather" conditions (open bars). (b) Percentage of missing wind lidar data for different wind speed classes, based on the mast measurements at the corresponding levels. For the 252 m wind lidar level, mast wind speed measurements at 200 m are taken.

## 5 Results

### 5.1 Data availability

The wind lidar was operating 99.4 % of the time, with the most significant downtime July 26-29 2019, due to a full internal storage issue of the wind lidar. In the following we consider data availability with respect to the uptime of the wind lidar. In Fig. 5(a) the overall availability of the QC 10-minute averaged wind data is shown by the filled bars, which ranges between 96.8 % and 98.4 %. The wind speed distribution of the 2 % to 3 % of missing wind lidar data is shown in Fig. 5(b). The lowest wind speed class (<4 m/s) shows the largest reduction in QC data, especially for the upper levels, while for moderate wind

speeds (8 m/s-16 m/s) the reduction is the least.

In Fig. 5(a) also the availability under "fair weather" conditions are given (open bars), which are very close to 100 %. Fair weather is defined here as no precipitation, visibility at 2 m in terms of meteorological optical range (MOR) more than 5 km and first cloud base height more than 1 km, which accounts for 58 % of the data. We will now take a closer look at the possible meteorological conditions that cause the decrease in QC data. Note that the meteorological data considered here are also 10

minute averaged.

In Fig. 6(a) and (b) the QC data availability is shown for different classes of precipitation intensity (as measured by the rain gauge) and presence of fog or low clouds (based on visibility measurements in the mast), respectively. The occurrences of the classes are given by the percentages between brackets. The different measuring heights are indicated by the colors. We notice that light precipitation, up to 0.1 mm/h, hardly affect the QC data availability. Only from an intensity of 1 mm/h onwards

we observe a significantly reduction, but mostly for the lower measuring heights. This is related to the height-dependent probe length. At the lower levels (short probe lengths) individual hydrometers can cause huge fluctuations in the return signal strength,





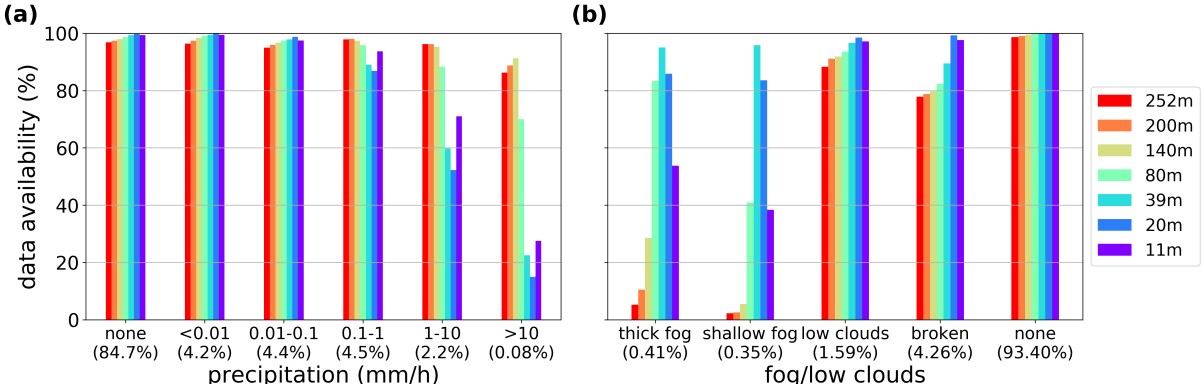

**Figure 6.** QC data availability of the wind lidar for (a) different classes of precipitation intensity and (b) presence of fog or low clouds (in absence of precipitation). The colors indicate the measuring heights (see legend). Percentages between brackets are the occurrences of the classes.

which can have a detrimental impact on the wind retrieval. At the upper levels (long probe lengths) individual hydrometers are not resolved.

It is well known that low clouds and fog can limit the wind lidar performance due to the attenuation of the laser light. Here
we have defined the fog/low clouds classes on basis on the visibility measurements in the A- and B-mast, which are performed at 7 levels from 2 m to 200 m (see Table 2). Events with precipitation are filtered out. The presence of fog (or clouds) at a certain level is triggered by visibility (MOR) less than 1 km. The (mutually exclusive) classes are:

- *thick fog*: fog at all levels;

- *shallow fog*: fog up to 80 m height (but not at all of the higher levels);

- *low clouds*: fog at 140 m and 200 m (but not at all of the lower levels);

- *broken*: fog at at least one level (but not fitting in one of the previous classes);

- *none*: no fog at any level.

We observe that fog in the lower 100 m (thick and shallow fog) has a detrimental impact on the QC data availability of the upper measuring levels. As fog is typically correlated with low wind speeds, this also explain the relative large reduction of
QC data for low wind speeds, as shown in Fig. 5(b). Interestingly, QC data availability at the moderate levels remain high, even under thick fog conditions. Clouds above 100 m do not have much impact. The reason why the low cloud class has more QC data for the upper measuring levels than the broken class might be due to enhanced backscatter from the cloud base compensating the attenuation below the clouds.





The analysis based on the visibility measurements in the mast is consistent with that of the first cloud base height derived

from the ceilometer[2]. We observed that from a CBH of 100 m or higher, the impact on amount of QC data is small, with the higher measuring heights being more affected, but still above 90 % even if measuring height is above CBH. Below a CBH of 100 m the amount of QC data is significantly reduced for measuring heights of 80 m and higher.

## 5.2 Data quality

We assess the data quality of the wind lidar by a comparison with the cup anemometers and wind vanes in the masts. Wind

measurements are sensitive to local obstacles, such a (rows of) trees, in particular for the lower levels. This limits the correlation between measurements at different locations. The distances between the wind lidar and the masts are relatively large, up to a few hundred meters (see Table 2). Therefore several measures have been taken to create a fair comparison. This includes considering only the nearest mast for the 10 m wind (D-mast), omitting the 20 m mast wind as this is measured by the further away B- and C-masts, and selecting only data from the 200°-250° wind sector, which has a long free stream (van Ulden and

Wieringa, 1996; Verkaik and Holtslag, 2007). The latter also circumvents the effect of the constructions on the RSS south to the wind lidar and trees on the east side of RSS (see Fig. 1(c) and Fig. 14(b)). Note that the selected wind sector overlaps with the prevailing wind direction (see Fig. 2(a)), and still 22 % of the data is present in the data quality analysis.

The following steps are taken to construct the "reference" mast wind data set:

1. Take the original Cabauw 10-minutes averaged, quality controlled wind speed and direction (10 m, 20 m 40 m, 80 m,
140 m and 200 m), based on measurements from the A-, B- and C-masts ("*original mast wind data set*").

2. Interpolate (cubic spline) each wind profile of the *original mast wind data set* to obtain a 11 m and 39 m mast wind speed and direction (forming the "*extended original mast wind data set*").

3. Take the 10-minutes averaged D-mast wind speed and direction (10 m).

4. Derive a 11 m D-mast wind speed by rescaling the 10 m D-mast wind speed by the *extended original mast wind data set*
11 m/10 m wind speed ratio.

5. Derive a 11 m D-mast wind direction by adding the difference between the 11 m and 10 m wind direction in the *extended original mast wind data set* to the 10 m D-mast wind direction.

6. Combine the 11 m D-mast wind with the *extended original mast wind data set* at 39 m, 80 m, 140 m and 200 m, omitting 20 m.

7. Select wind sector between 200° and 250°, based on 10 m wind from D-mast.

8. Select wind speeds larger than 0.5 m/s.

---

[2]The Lufft CHM15K ceilometer was operating at firmware v0.747 or higher in combination with a "low cloud detection mode" during the measurement campaign.





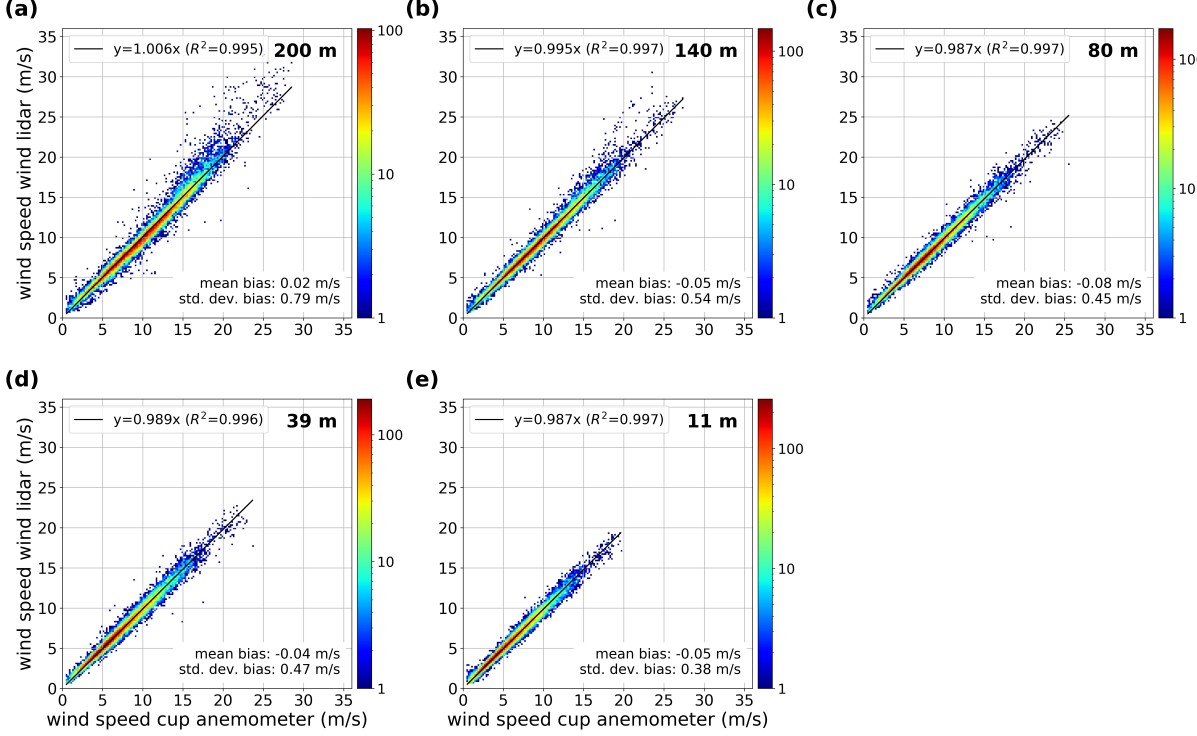

**Figure 7.** Wind speed comparison between wind lidar and mast data for the different heights: (a) 200 m, (b) 140 m, (c) 80 m, (d) 39 m and (e) 11 m. The results of a linear regression analysis (with and without offset), and the mean bias and standard deviation in the bias are indicated in each panel.

In Appendix A a comparison of the results from different wind sectors are provided. For the lower levels (up to 39 m) differences are indeed observed, with the free stream wind sector (200°-250°) showing the "best" comparison. For the upper levels (from 80 m onwards) the main results are mostly independent from the chosen wind sector. However, to be consistent

we have applied the wind sector selection to all levels.

### 5.2.1 Horizontal wind speed

The horizontal wind speed data of the wind lidar and the mast measurements are compared for five heights. Scatterplots are presented in Fig. 7. Linear regression fits are shown and the correlation coefficient $R^2$ is provided. In addition, the mean bias and standard deviation are given. Bias is defined as the wind lidar wind speed minus mast wind speed. For visualization

purposes the scatterplots are presented as density plots with finite bin sizes, with a logarithmic color scale. The fits and biases are related to the individual data points. For the linear regression we find the slope ranging from 0.99 to 1.00 with $R^2$ better than 0.995. The mean bias is between -0.08 m/s and 0.02 m/s, and the standard deviation between 0.4 m/s and 0.8 m/s. These results are considering the full wind speed range.

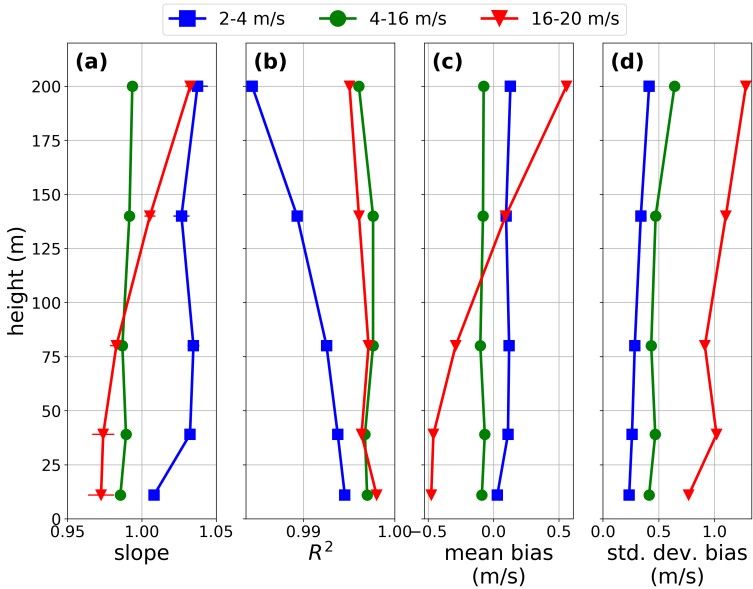

**Figure 8.** Profiles of linear regression analysis results and biases in the wind speed, applied to three different wind speed classes: 2-4 m/s (blue squares), 4-16 m/s (green circles) and 16-20 m/s (red triangles). Panels (a) and (b) shows the parameters for a linear regression without offset (slope and $R^2$), and panels (c) and (d) the mean bias and standard deviation of the bias. The error bars in panel (a) indicate the standard uncertainties of the parameter estimates (often smaller than symbol).

Within the wind energy industry wind lidars are typically validated only between 4 m/s and 16 m/s, while from a meteoro-
logical point of view lower and higher wind speeds are also of interest. In Fig. 8 results of linear regression and the biases for
three different wind speed classes (2-4 m/s, 4-16 m/s, 16-20 m/s) are shown. The results for the 4-16 m/s class: slope ranging
from 0.99 to 1.00 with $R^2$ better than 0.996; mean bias between -0.10 m/s and -0.07 m/s, standard deviation between 0.4 m/s
and 0.6 m/s. These results are similar to the ones shown in Fig. 7 based on the full wind speed range.

For lower wind speeds (2-4 m/s) the slope deviates more from 1 and the correlation is smaller compared to the 4-16 m/s
class. Here the mean bias is slightly positive, between 0.03 and 0.13 m/s. A positive bias for low wind speeds has been reported
(Courtney et al., 2008), which was related to the inability of the homodyne wind lidar to measure zero Doppler-shift. However,
we have found that the wind lidar reported wind speeds down to 0.6 m/s, much lower the minimum wind speed in this class. We
note that the accuracy of the cup anemometer is 0.1 m/s in this wind speed region, which is on the same order as the observed
mean bias.

For higher wind speeds (16-20 m/s) larger deviations are observed, and the bias varies from -0.5 m/s to 0.5 m/s with
measuring height. For even higher wind speeds (>20 m/s), visual inspection of Fig. 7 indicates a positive bias and large
scatter for 140 m and 200 m. This feature was already noted in the example presented in Fig. 4. As discussed above, possible
non-linearity of the cup anemometer calibration above 20 m/s, which indeed would give a positive bias, only will give a





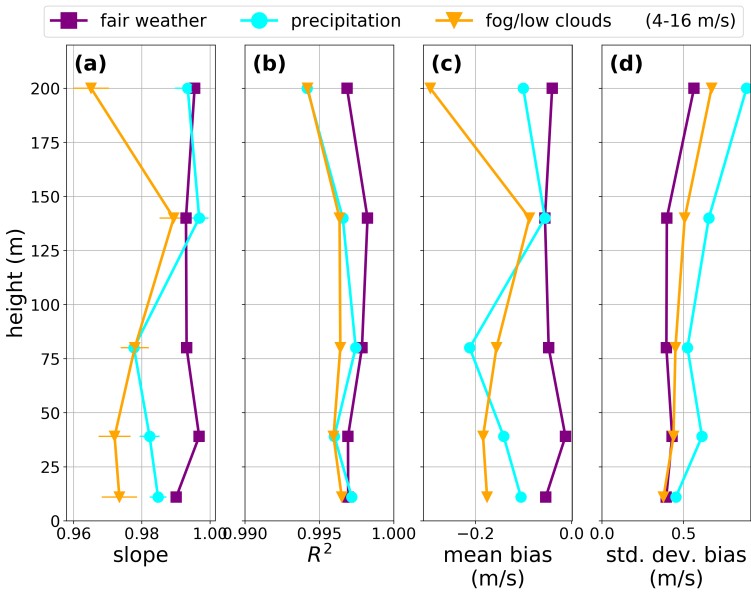

**Figure 9.** Profiles of linear regression analysis results and biases in the wind speed, applied to three different meteorological conditions: fair weather (purple squares), precipitation (cyan circles) and fog/low clouds (orange triangles). Here wind speed is bounded to the 4-16 m/s range. Panels (a) and (b) shows the parameters for a linear regression without offset (slope and $R^2$), and panels (c) and (d) the mean bias and standard deviation of the bias. The error bars in panel (a) indicate the standard uncertainties of the parameter estimates (often smaller than symbol).

gradual deviation with increasing wind speed and cannot explain these large differences between the wind lidar and the mast

measurements.

The co-located meteorological observations allow to verify the QC wind lidar data for different weather conditions. In Fig. 9 results of linear regression and the biases are shown for "fair weather", "precipitation" and "fog/low clouds" conditions. "Fair weather" is defined as above (no precipitation, MOR>5 km at 2 m and first cloud base height more than 1 km), for "precipitation" a threshold of 0.1 mm/h is taken, and "fog/low clouds" requires at least one mast level with MOR<1 km, while

precipitation events are filtered out. Here wind speed is bounded to the 4-16 m/s range for a more fair comparison, recognizing that different weather conditions may be connected to different typical wind speeds.

The fair weather condition gives overall the best results, while the possible impact of precipitation or fog/low clouds on the data quality is small. Most notable is a more negative mean bias at most measuring heights, up to -0.3 m/s at 200 m for fog/low clouds.

**5.2.2 Wind direction**

The wind direction data of the wind lidar and the mast measurements are compared for five heights. Scatterplots are presented in Fig. 10. The selected wind sector is between 200° and 250°, as measured at 10 m. The mean bias and standard deviation are





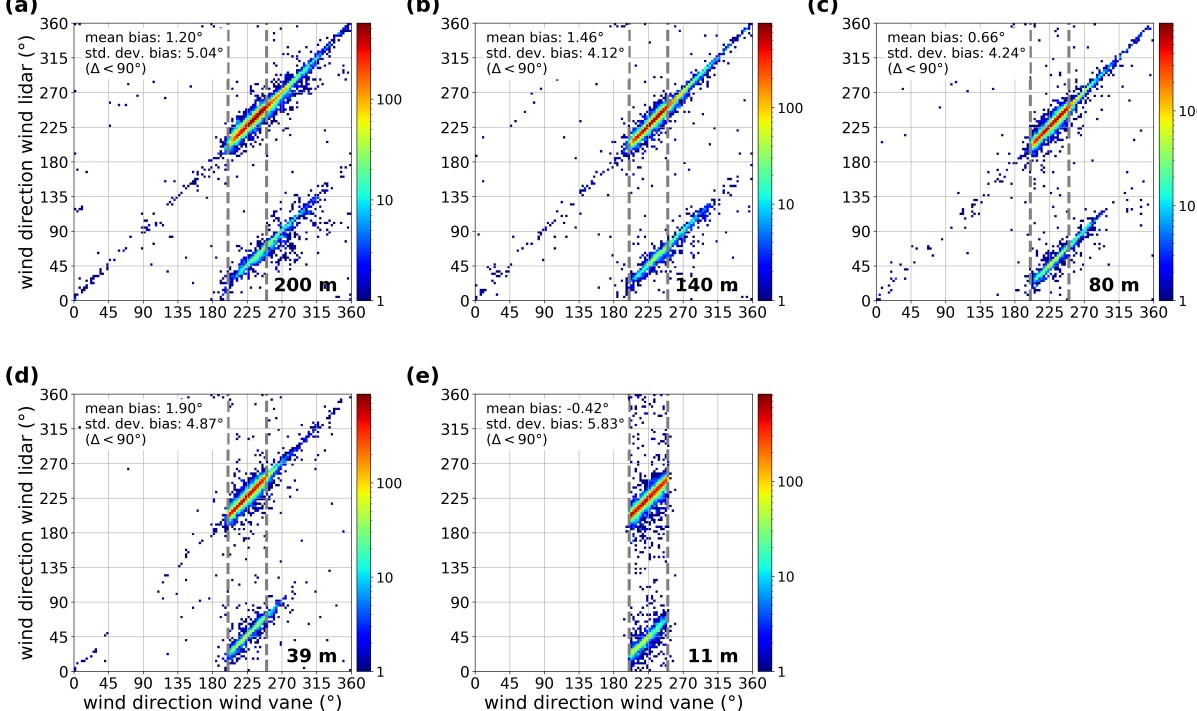

**Figure 10.** Wind direction comparison between wind lidar and mast data for the different heights: (a) 200 m, (b) 140 m, (c) 80 m, (d) 39 m and (e) 11 m. The mean bias and standard deviation in the bias are indicated in each panel, taking into account only data for which the deviation is less than 90°. Note that the selected wind sector of 200°-250° is based on the 10 m wind.

indicated. Bias is defined as the wind lidar wind direction minus mast wind direction. For visualization purposes the scatterplots are presented as density plots with finite bin sizes, with a logarithmic color scale. The biases are related to the individual data 265 points. The presence of data points around $\pm180°$ away from the $y = x$ line will be discussed in Sect. 5.2.3. Here only the wind lidar data for which the difference with the mast is less than 90° is taken into account. We find values of the mean bias ranging from -0.4° to 1.9°, which is within the combined accuracy of the wind vanes and the alignment of the wind lidar. The standard deviation in the bias is between 4° and 6°.

In Fig. 11(a)-(b) results of the bias for the three different wind speed classes (2-4 m/s, 4-16 m/s, 16-20 m/s) are shown. 270 The variation in the mean bias is within the combined accuracy of the wind vanes and the alignment of wind lidar. When considering wind speeds above 4 m/s, the standard deviation is 3° or less. For low wind speeds the standard deviation in the bias is much larger, which is mainly a property of the wind field itself, rather than the instruments.

In Fig. 11(c)-(d) the biases are shown for "fair weather", "precipitation" and "fog/low clouds" conditions (as defined above), for which the wind speed is bounded to the 4-16 m/s range. Again, the variation in the mean bias is within the combined 275 accuracy of the wind vanes and the alignment of wind lidar. Precipitation and fog/low clouds show a slight increase in the standard deviation towards higher levels, up to 7°.





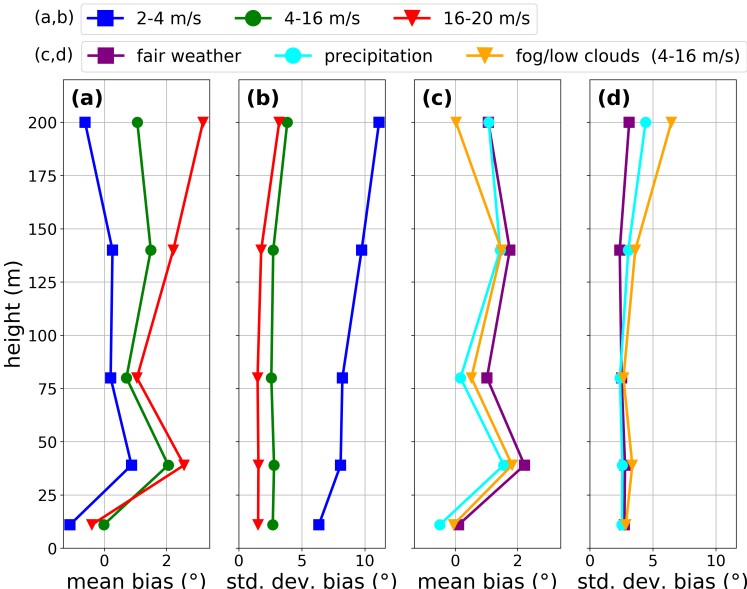

**Figure 11.** Profiles of the biases in the wind direction for different wind speed classes and meteorological conditions. Panels (a) and (b) show the mean bias and standard deviation of the bias applied to three different wind speed classes: 2-4 m/s (blue squares), 4-16 m/s (green circles) and 16-20 m/s (red triangles). Panels (c) and (d) show the mean bias and standard deviation of the bias applied to three different meteorological conditions (and wind speed is bounded to the 4-16 m/s range): fair weather (purple squares), precipitation (cyan circles) and fog/low clouds (orange triangles).

We note that the method of deriving the 10-minute wind direction averages differ between the wind lidar (wind vector averaging) and the mast measurements (unit vector averaging), which in principle could lead to slightly different results. However, we expect significant effects only for periods of very large wind direction variations, such as low wind speeds or convective situations, which are for instance unlikely to be present in the 4-16 m/s wind speed class.

### 5.2.3  180° ambiguity

The ZephIR 300 instrument is based on homodyne detection, meaning that only the absolute value of the Doppler-shift is measured. As a result there is a 180° ambiguity in the measured wind direction. To solve this issue the ZephIR 300 includes an attached meteo station, which contains a sonic anemometer to measure the wind direction just above the instrument, i. e. a height of 1 m. This information is used in the instrument's internal algorithm to determine the true wind direction of the wind lidar measurements. Still, the occurrence of incorrectly assigned wind direction events is possible, resulting in part of the wind direction data that is off by 180°. This is most likely in situations of very low wind conditions, in which wind direction is not very well defined, but can also be caused by nearby obstructions that alter the wind flow at the location of the meteo station.

The incorrectly assigned wind direction events can be observed 180° off from the $y = x$ line in Fig. 10. For many applications it is sufficient to correct the wind direction data offline. However, for real-time applications, such as nowcasting, it would be



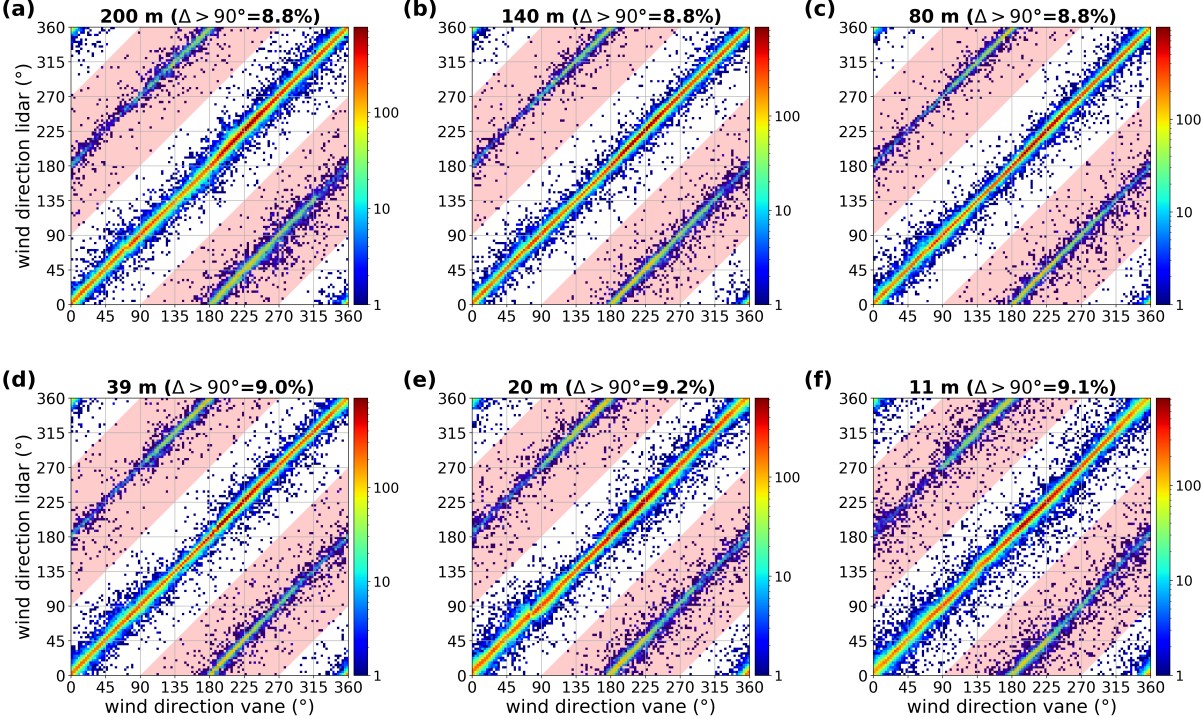

**Figure 12.** Wind direction comparison between wind lidar and mast data for the different heights: (a) 200 m, (b) 140 m, (c) 80 m, (d) 39 m, (e) 20 m and (f) 11 m. Compared to Fig. 10, here all wind directions the 20 m level are included. The regions of incorrectly assigned wind direction are indicated in red; the percentage of data in those areas are indicated on top of each panel.

desirable to (1) minimize their occurrence, and (2) correct in real-time. Here we have compared two positions of the meteo station and considered a correction scheme based on mast wind direction information at a single level.

In order to quantify the occurrence of incorrectly assigned wind direction we consider events for which the absolute differ-ence between the wind lidar and the wind vane (at the same measuring height), denoted $\Delta$, is more than $90°$ ($\Delta > 90°$)[3]. This

is motivated by the observation of clearly separate "groups" of data around the $y = x$ and $y = x \pm 180°$ lines (i. e. the standard deviation in the bias is much smaller than $180°$). We now take data from all wind directions, not only free flow stream, and included 20 m height, as accuracy is less crucial here. Fig. 12 shows again scatterplots of the wind direction data, but now including all wind directions and the 20 m level. The $\Delta > 90°$ events are located in the red colored regions; their percentages are shown above each panel, which is about 9 %, with little variation over the heights.

In Fig. 13(a) the percentages of $\Delta > 90°$ events are also shown for the different heights. In addition, results for wind speeds above 4 m/s are shown, either related to wind speed measured at 10 m or at the corresponding height of the wind lidar measurements. First of all, omitting low wind speeds leads to a reduction of the $\Delta > 90°$ occurrences. Second, their occurrences are dependent on the wind speed near the instrument, rather than the measuring height of the wind lidar. This

---

[3]after folding the wind direction differences in the -180° to 180° range





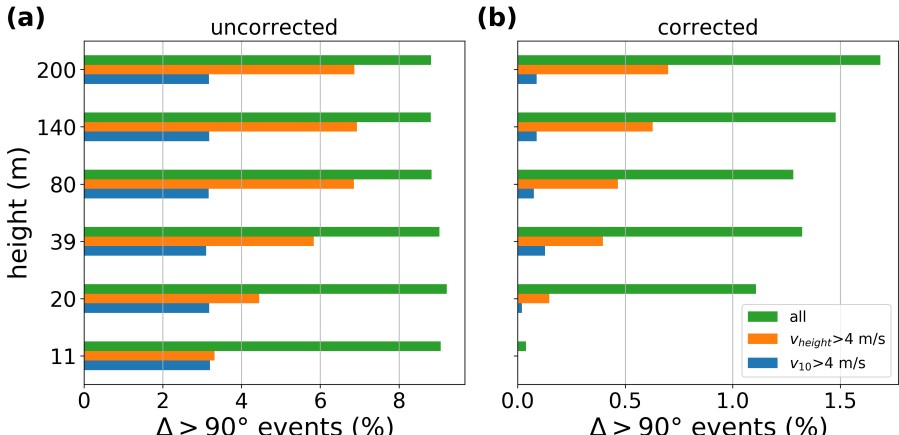

**Figure 13.** Panel (a): histogram of the $\Delta > 90°$ events for the different measuring heights, based on either all events (green), wind speeds at the measuring height larger than 4 m/s (orange) and 10 m wind speeds than 4 m/s (blue). Panel (b): same as Panel (a) but after application of the correction scheme (see text), based on information from the 10 m mast wind vane.

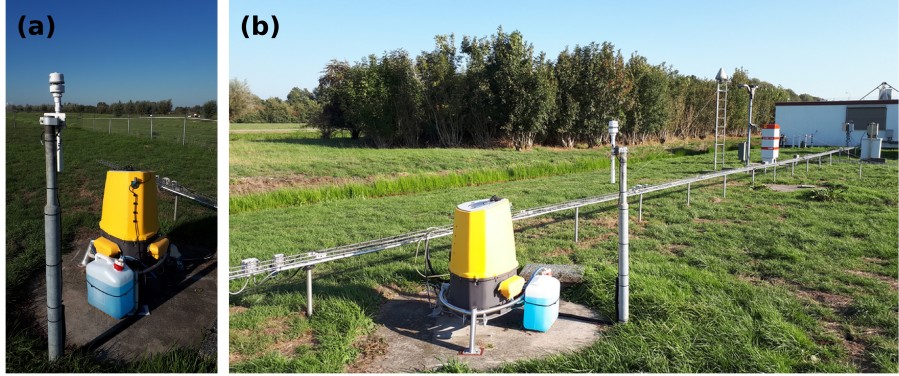

**Figure 14.** Photos of (a) the meteo station of ZephIR 300 instrument located on a separate pole and (b) a view to the South-East direction, showing building on RSS and nearby trees.

explains the moderate reduction at the highest levels, for which wind speeds above 4 m/s still can be connected with much
lower wind speeds near the surface.

The standard location of the meteo station is directly on top of the ZephIR 300, as can be seen in Fig. 1(b). However, it is possible (and in some cases recommended by the manufacturer) to install the meteo station separately from the wind lidar, for instance if the wind lidar is enclosed within (open) fences. We have relocated the meteo station after six months of the measurement campaign to a separate pole, at a height of 1.5 m (see Fig. 14(a)), for the remaining 1.5 years.

In Fig. 15 we show the $\Delta > 90°$ occurrence for the lowest measuring height (11 m) for different wind sectors, separating the data regarding the location of the meteo station. We see that the $\Delta > 90°$ occurrence depends on the wind direction, being





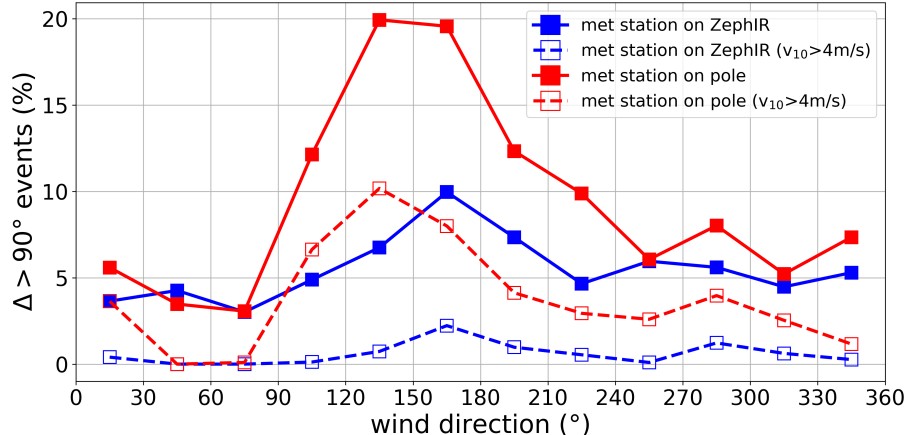

**Figure 15.** The $\Delta > 90°$ occurrence for the lowest measuring height of 11 m for different wind sectors (binsize $30°$), separating the data regarding the location of the meteo station and including a wind speed threshold of 4 m/s (as measured at 10 m).

much more prominent for Southerly wind. This can be explained by the presence of buildings and large instruments at RSS and trees to the east side of RSS that disturb the wind flow at the meteo station (see Fig. 14(b)). Again, when considering only wind speeds larger than 4 m/s (as measured at 10 m) $\Delta > 90°$ drops significantly. We cannot explain the observation that $\Delta > 90°$

has occurred more often for the meteo station on the separate pole than directly on the ZephIR 300 instrument.

The severity of how the $180°$ issue is present in the wind lidar measurements depends on both wind speed, and wind direction through the characteristics of its surroundings. The low height of the meteo station means that low wind conditions can prohibit reliable wind direction measurements and being more sensitive to disturbance in the wind flow by nearby objects. The height increase from 1.0 m to 1.5 m did not lead to a decrease of $\Delta > 90°$.

With additional (real-time) information about the actual wind direction, one can correct the $\Delta > 90°$ events. In case such information is only available at a single height, one can compare this with a corresponding or nearby height of the ZephIR 300, and each time $\Delta > 90°$ for this height change the ZephIR 300 wind direction data for *all* heights by $180°$. In Fig. 13(b) we show the result of such a correction on basis of the 10 m D-mast wind data. While the largest reduction is for the nearby height (by definition $\Delta > 90°$ will be absent when these heights are equal), also the reduction for the other height is significant, with

$\Delta > 90°$ drops from 9 % to well below 2 %. When considering wind speeds above 4 m/s $\Delta > 90°$ is further reduced well below 1 %, depending on which height the wind speed threshold is taken. The success of this correction scheme depends on the size of the natural wind veer between mast height and (highest) wind lidar measuring level, which should be well below $90°$.





## 6 Conclusions

We have conducted a two-year measurement campaign of the ZephIR 300 vertical profiling CW focusing wind lidar at the Cabauw site. We have studied the (height-dependent) data availability of the wind lidar under various meteorological conditions and the data quality of 10-minute averaged horizontal wind speed and wind direction via a comparison with in situ wind measurements at several levels in the 213-m tall meteorological mast.

We find an overall availability of QC data of 97 % to 98 %, where the missing part is mainly due to precipitation events ex-
ceeding 1 mm/h or fog or low clouds below 100 m. Precipitation affects mostly the lower measuring levels, fog and low clouds the upper ones. The mean bias in the horizontal wind speed is within 0.1 m/s with a high correlation between the mast and wind lidar measurements, although under some specific conditions (very high wind speed, fog or low clouds) larger deviations are observed. The mean bias in the wind direction is within 2°, which is of the same order as the combined uncertainty in the alignment of the wind lidars and the wind vanes.

The 180° error in the wind direction output occurs about 9 % of the time, which is reduced when omitting low wind speeds. This percentage depends strongly to amount of possible wind flow disturbance near the attached meteo station. A correction scheme based on data of an auxiliary wind vane at a height of 10 m is applied, leading to a reduction of the 180° error below 2 % (or even well below 1 % when considering wind speeds above 4 m/s). This scheme can be applied in real-time applications in case a nearby, freely exposed, mast with wind direction measurements at a single height is available.

In this work we have focused on the most commonly used output for meteorology and wind energy purposes: 10-minute averaged horizontal wind speed and wind direction. However, the wind lidar output also contains minimum, maximum and standard deviation of the horizontal wind speed and turbulence intensity, which are likely to be more sensitive to fundamental differences between the cup anemometer and the wind lidar (see e. g. Sathe et al. (2011); Suomi et al. (2017)) and therefore an intercomparison is recommended for those parameters as well. The vertical wind speed output might be compared with the
sonic anemometers that are present at the 60 m, 100 m and 180 m mast levels.

*Code and data availability.* Datasets and software codes are available at http://doi.org/10.5281/zenodo.3966868. Note that Cabauw tower and surface meteorological data are also available via https://dataplatform.knmi.nl/.

## Appendix A: Wind sector selection

For the wind speed and wind direction intercomparison the long free flow wind sector between 200° and 250° is selected. Here
we show some results on the wind speed intercomparison for the full range of wind directions. In Fig. A1 the relative deviation between the wind lidar and the mast wind speed measurements is shown as function of the wind direction (at 10 m), in which the data is collected in bins of 20°, for wind speeds ranging between 4 m/s and 16 m/s. For the upper four heights, no wind direction dependence of observed, whereas for the lower two heights a clear modulation of the relative deviation is visible, with a negative deviation between 100° and 150° and positive deviation between 250° and 300°. We explain this behavior by

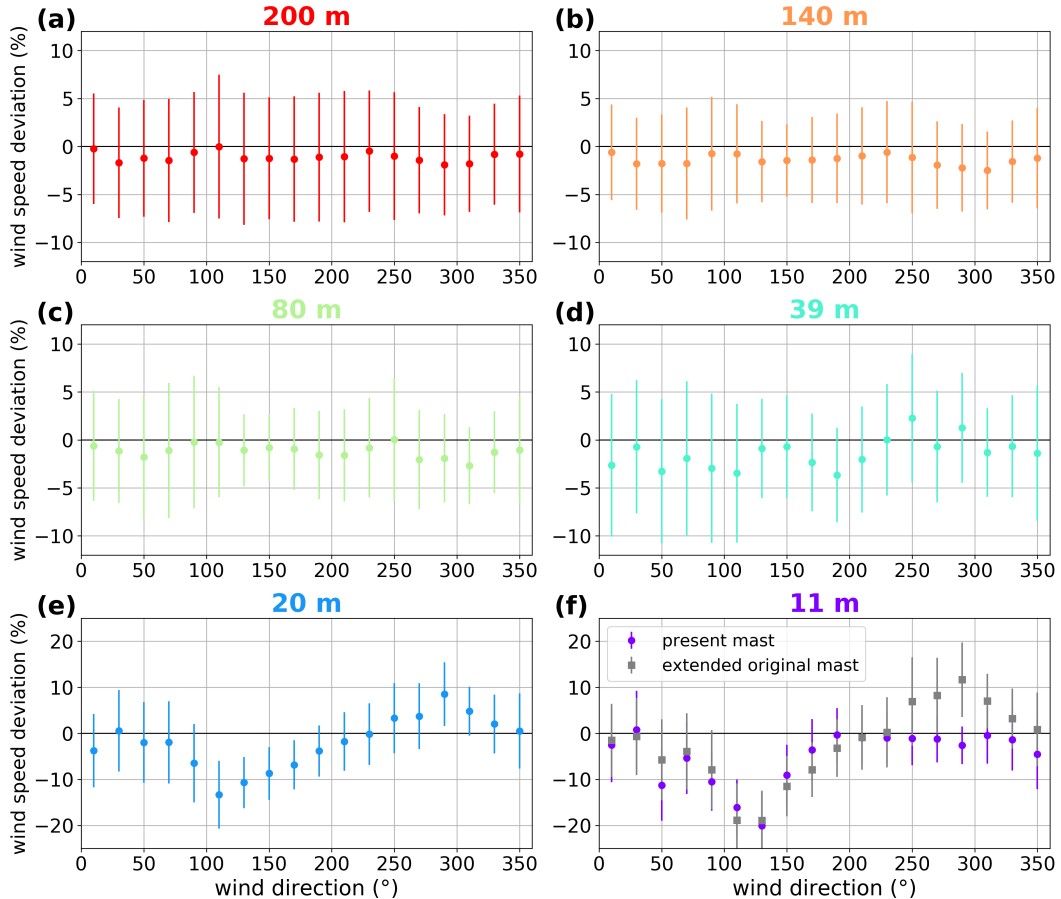

**Figure A1.** Relative deviation between the wind lidar and the mast wind speed measurements as function of wind direction (at 10 m) for the different heights: (a) 200 m, (b) 140 m, (c) 80 m, (d) 39 m, (e) 20 m and (f) 11 m. The data points represent the bin average and standard deviation (wind direction binsize is $20°$). Wind speeds are bounded to the 4-16 m/s range. Note the different scales.

the presence of flow obstruction at the remote sensing site and neighboring trees SE of the wind lidars (for $100°$-$150°$), and neighboring trees NW of the C-masts (for $250°$-$300°$), in combination with the relatively large distance between the wind lidar and the reference masts. This last feature is absent in the present 11 m mast data, because the D-mast is taken instead of the C-mast.

In Fig. A2 a comparison between the linear regression results and the biases for different wind sectors is made, based on the
present reference mast wind data set, and again for wind speeds ranging between 4 m/s and 16 m/s. The results for the slope (panel (a)) and the mean bias (panel (c)) clearly shows sensitively for the wind direction for the lower levels (up to 39 m), whereas for the upper levels (from 80 m onwards) the results of the different wind sectors overlap. For the full profile the $200°$-$250°$ wind sector provides the best results. The results for $R^2$ (panel (b)) and the standard deviation (panel (d)) show some variation among the different wind sectors over the full profile. For $R^2$ we note that the span is very narrow (well within





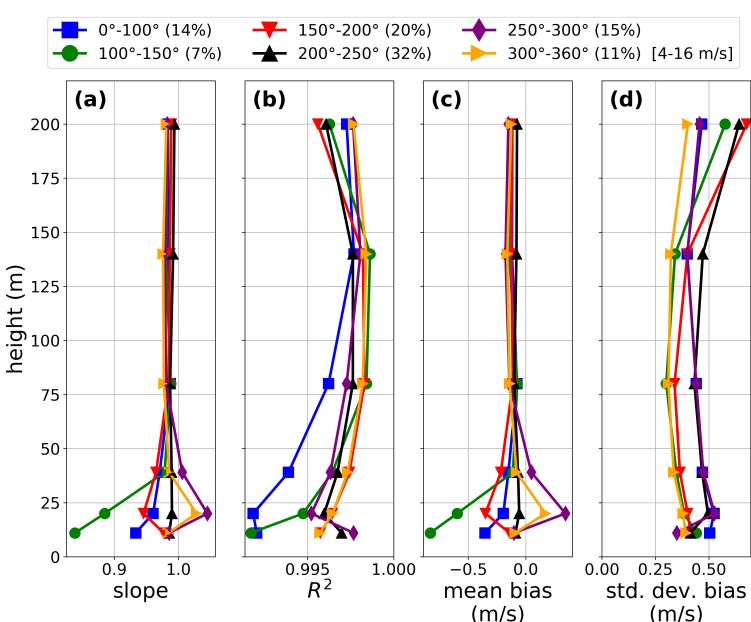

**Figure A2.** Profiles of linear regression analysis results and biases in the wind speed, applied to different wind sectors, where the wind speed is bounded to the 4-16 m/s range. Panels (a) and (b) shows the parameters for a linear regression without offset, and panels (c) and (d) the mean bias and standard deviation of the bias. The percentages in the legend indicate the occurrences of the different wind sectors.

0.005) for the upper levels. For the standard deviation the differences might be linked to the distinct wind speed distributions of the different wind sectors (see Fig. 2). For instance, the mean wind speed (within the 4-16 m/s class) is the highest for the 150°-200° and 200°-250° wind sectors (9.8 m/s at 200 m), and the lowest for the 300°-360° wind sector (7.8 m/s at 200 m).

*Author contributions.* SK was responsible for the wind lidar measurements, performed the data analysis and wrote the original draft. FB was responsible for the mast wind measurements and post-processing. All co-authors contributed to refining the manuscript text.

*Competing interests.* The authors declare that they have no conflict of interest.

*Acknowledgements.* We acknowledge Rijkswaterstaat Maritiem Informatievoorziening Service Punt (RWS-MIVSP) for supporting this measurement campaign and providing a ZephIR 300M instrument. We also acknowledge Alfons Driever (KNMI) for technical support.



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
