# Peer review of "A two-year intercomparison of CW focusing wind lidar and tall mast wind measurements at Cabauw"

_Atmospheric Measurement Techniques, 2020_

## Referee Comment (RC1) · Anonymous Referee #1 · 24 Nov 2020

This is an excellent paper investigating the long term performance and measurement capability of a commercial Doppler lidar under the range of measurement conditions encountered over a two-year period. The data analysis and presentation of results is quite meticulous, with critical areas important for application of this lidar identified, examined and discussed. I find the paper for the most part acceptable as submitted, but here include a few minor comments which the editor and authors can optionally consider.

Line 25: Coherent lidars also measure the shift of the frequency spectrum, albeit by a different methodology.

[Figure]

Line 60: It should be noted that changing the focal length also changes the depth of focus (i.e., probe length).

Line 65: It would have been interesting to find cases over the two year period where clouds existed beyond the minimum range to investigate performance of the cloud removal algorithm.

Line 192: It seems possible in the broken cloud case that the presence of clouds beyond the maximum range might be leading to more QC failures. Broken clouds might be expected to be higher and more likely to contribute to this effect.

Line 257: Higher biases at longer range under low cloud/fog conditions could be due to asymmetric range weighting, where the signal at the more distant range of the probe volume has been attenuated relative to that at the closer, resulting in the effective measurement distance being closer to lidar than the focal range.

Line 269-272: These are quite good results.

Figure 13 (caption): Seems like the caption is missing the word "greater" before "than 4 m/s" in line 2.

---

## Referee Comment (RC2) · Anonymous Referee #2 · 1 Dec 2020

**A two-year intercomparison of CW focusing wind lidar and tall mast wind measurements at Cabauw**

Steven Knoop[1], Fred C. Bosveld[1], Marijn J. de Haij[1], and Arnoud Apituley[1]

[1]Royal Netherlands Meteorological Institute (KNMI), Utrechtseweg 297, 3731 GA De Bilt, The Netherlands

**Correspondence:** Steven Knoop (steven.knoop@knmi.nl)

**Abstract.** A two-year measurement campaign of the ZephIR 300 vertical profiling continuous-wave (CW) focusing wind lidar has been carried out by the Royal Netherlands Meteorological Institute (KNMI) at the Cabauw site. We focus on the (height-dependent) data availability of the wind lidar under various meteorological conditions and the data quality through a comparison with in situ wind measurements at several levels in the 213-m tall meteorological mast. We find an overall availability of quality controlled wind lidar data of 97 % to 98 %, where the missing part is mainly due to precipitation events exceeding 1 mm/h or fog or low clouds below 100 m. The mean bias in the horizontal wind speed is within 0.1 m/s with a high correlation between the mast and wind lidar measurements, although under some specific conditions (very high wind speed, fog or low clouds) larger deviations are observed. The mean bias in the wind direction is within $2°$, which is on the same order as the combined uncertainty in the alignment of the wind lidars and the mast wind vanes. The well-known $180°$ error in the wind direction output for this type of instrument occurs about 9 % of the time. A correction scheme based on data of an auxiliary wind vane at a height of 10 m is applied, leading to a reduction of the $180°$ error below 2 %. This scheme can be applied in real-time applications in case a nearby, freely exposed, mast with wind direction measurements at a single height is available.

**1 Introduction**

Atmospheric motion and turbulence are essential parameters for weather and topics related to air quality. Therefore, wind profile measurements play an important role in atmospheric research and meteorology. One source of ground-based wind profile data are Doppler wind lidars, which are active, laser-based remote sensing instruments that measure wind speed and wind direction up to a few hundred meters or even a few kilometers. Like traditional radar wind profilers, Doppler wind lidars typically cover the atmospheric boundary layer very well and thereby complement other sources of wind information, such as in-situ measurements at surface stations, weather radars, aircraft observations and satellite instruments. An extensive and comprehensive description of the history and fundamentals of wind lidars can be found in Henderson et al. (2005).

[Figure]

**Figure 1.** (a) Map of the Netherlands, indicating the location of measurement site near Cabauw (51.971° N, 4.927° E). (b) Photo of the ZephIR 300 wind lidar instrument, with the 213-m tall A-mast visible the background (view in NW direction). (c) Overview of the locations of the masts, the ZephIR wind lidar and other relevant instruments: rain gauge (RG), ceilometer (CBH) and present-weather sensor (PWS); the inset shows the area around the site (images from Google maps).

Within the wind energy industry short-range vertical-profiling wind lidars are popular and well-accepted instruments, typi-cally used for wind resource assessment and wind turbine power curve validation (Mikkelsen, 2015). These wind lidars operate
25   at a laser wavelength around 1.5 $\mu$m, and the backscatter is mainly caused by aerosols. 
[revised manuscript text omitted]

---

## Author Comment (AC1) · 2 Dec 2020

We thank Referee #1 for his or her compliments and interesting comments on our paper. We now will reply to the comments one by one. Intended changes to the manuscript are explicitly mentioned.

(1) Line 25: "Coherent lidars also measure the shift of the frequency spectrum, albeit by a different methodology."

Probably this boils down to the question whether measuring a beat-signal is the same as measuring a shift of the frequency spectrum. However, what we had in mind is that

in direct detection the frequency spectrum of the backscattered light is detected, from which the frequency shift with respect to the emitted light is determined. In coherent detection it is not the frequency spectrum of the backscattered light that is measured, but rather a beat-signal between the local oscillator (which is related to the emitted light via a frequency offset, which also can be zero).

We propose the following minor change: "Direct detection wind lidars measure the frequency spectrum of the return signal", omitting "the shift of", which makes more apparent the difference between the two techniques.

(2) Line 60: "It should be noted that changing the focal length also changes the depth of focus (i.e., probe length)."

A statement on the range dependent probe length was already present in the next paragraph (lines 64-65). We don't think it is necessary to make a change.

(3) Line 65: "It would have been interesting to find cases over the two year period where clouds existed beyond the minimum range to investigate performance of the cloud removal algorithm."

In our comparison of the wind lidar and mast in situ data we have considered the effect of fog and low clouds (see Figure 9 of the manuscript), but we didn't specifically considered the case of only clouds below 10 m. We can select cases on basis of the visibility sensors, with the condition of fog at 2m, but no fog at all the other levels (starting at 10m). The result of the intercomparison, including this class of clouds, is included here as Fig. 1. The "fog/low clouds" class is as defined in the paper, the new "fog below 10m" as given here. On a first glimpse the results look very good, better than the "fog/low clouds" class. However, one has to keep in mind that here wind speeds between 4 m/s and 16 m/s are considered, and that no information on the vertical thickness of the fog layer is available (rather than MOR<1000m at 2m and MOR>1000m at 10m and upwards). Thus the "fog below 10m" could include very thin and patchy cloud layers. To select events with a persistent fog layer below 10m,

[Figure]

one needs to analyze the visibility measurements on a sub-10 minute timescale and probably lower the upper limit for the MOR. So while we agree that the point raised by the referee is interesting, a thorough analysis of it is beyond the scope of this work.

(4) Line 192: "It seems possible in the broken cloud case that the presence of clouds beyond the maximum range might be leading to more QC failures. Broken clouds might be expected to be higher and more likely to contribute to this effect."

Indeed, the way we have constructed the "low cloud" class and "broken cloud" class leaves open the possibility that in the latter case there are more clouds also above the range provided by the mast (or the wind lidar). However, the observations with the ceilometer showed that a first cloud base height above 100m has little impact on the data availability, and therefore we don't expect this property of the broken cloud class to be the cause of smaller data availability.

(5) Line 257: "Higher biases at longer range under low cloud/fog conditions could be due to asymmetric range weighting, where the signal at the more distant range of the probe volume has been attenuated relative to that at the closer, resulting in the effective measurement distance being closer to lidar than the focal range."

This is indeed a possible reason for the observed negative bias for the fog/low clouds class. We will include this explanation in the paper.

(6) Line 269-272: "These are quite good results."

Thanks for the comment.

(7) Figure 13 (caption): "Seems like the caption is missing the word "greater" before "than 4 m/s" in line 2."

Thanks, indeed a word (either "larger" or "greater") was missing in front of the second "than 4 m/s" in this caption, and this will be corrected.
* * *
[Figure]

[Figure]

**Fig. 1.** Same as Figure 9 of the manuscript, but including the class "fog below 10m", with the condition of MOR<1000m at a height of 2m, and MOR>1000m at all other heights (starting at 10m).

---

## Author Comment (AC2) · 3 Dec 2020

We thank the referee for raising the issue of thermal expansion of the transceiver that could lead to deviations in the position of the focus, and therefore lead to a seasonal dependent bias in the ZephIR wind speed measurements. We will investigate a possible temperature effect on the intercomparison and if we find a significant effect we will include this in the manuscript.

The instrument output includes an internal temperature measured close to the window and we have data on the air temperature from the co-located automatic weather station as well as the meteo station attached to the instrument. This means we can do more

than a seasonal analysis.

Regarding the comments in the supplement:

(1) "Doppler wind lidars typically cover the atmospheric boundary layer very well and thereby complement other sources of wind information, such as in-situ measurements at surface stations, weather radars, aircraft observations and satellite instruments."

Referee #2: This is not correct. Direct detection doppler wind technique can cover a much higher range. https://journals.ametsoc.org/bams/article/86/1/73/58297 https://amt.copernicus.org/articles/6/3349/2013/

Response: We are well aware of wide variety of wind lidar instruments. The example of AEOLUS is given in our manuscript that was published online September 30, 2020, on line 25 (Referee #2 unfortunately is considering an earlier version of our manuscript, which was not published online). However, we would not call the AEOLUS wind lidar "typical". Furthermore, we have stated that those instruments "cover" the boundary-layer, which does not automatically imply that that is also the limiting range.

(2) "These wind lidars operate at a laser wavelength around 1.5 mum, and the backscatter is mainly caused by aerosols. For national meteorological services, like the Royal Netherlands Meteorological Institute (KNMI), data sets measured by these instruments can be valuable for model validation, while real-time access opens the possibility of data assimilation in operational numerical weather prediction (NWP) models and nowcasting purposes. For these applications it is of utmost importance to know the meteorological conditions in which the instruments are able to provide reliable data or not."

Referee #2: It would be better to add a description between the different lidar techniques, e.g. direct detection, heterodyne. Also specify the used medium (aerosols, molecules)

Response: In our manuscript that was published online September 30, 2020, we had

extended the general discussion on Doppler wind lidar.

(3) "The height range is 10-200 m above the instrument, although up to 300 m can be selected in the software."

Referee #2: doesn't matter what you can chose in the sofware. The instrument works when aerosols are available

Response: We agree that this instrument requires aerosols to measure wind. This is also explicitly mentioned in the introduction. However, for our location, and up to a height of 300m, enough aerosols are always present. In fact, we are not aware of limited data availability due to lack of aerosols for short-range wind lidars (such as the ZephIR 300 or the Windcube v2). Could the referee provide publications indicating situations of too low aerosol signal for these short-range wind lidars?

(4) "As a result, the wind lidar can resolve the wind profile better than the mast."

Referee #2: this is pretty obvious, considering also the different costs between lidar and mast

Response: We don't understand the point the referee wants to make here. If the referee thinks the lidar costs are higher than the masts, then he or she underestimates the cost of mast wind measurements (including the mast infrastructure, maintenance, calibration).

(5) "The wind lidar and mast measurements are in close agreement for most of the levels, with the exception of some part of the day where the wind speed was around 25 m/s or higher, occurring mostly at 140 m and 200 m"

Referee #2: why those discrepancies at higher wind speeds ?

Response: This is indeed a very interesting observation. Unfortunately, we have no explanation on the bias for higher wind speeds at those levels. As we wrote in Section 5.2.1, we don't think the discrepancy is due to the cup anemometers.

---

## Author Comment (AC3) · 9 Dec 2020

The referee has raised the issue of thermal expansion of the transceiver that could lead to deviations in the position of the focus, and therefore lead to a seasonal dependent bias in the ZephIR wind speed measurements. Fig. 1 shows the analysis for the two summers and winters of our measurement campaign, where panel (c) indeed indicate a summer-winter difference of about 0.1-0.2 m/s for the highest levels. Fig. 2 shows the analysis for different air temperature classes (on 10-min. basis). All panels do seem to show a trend in temperature. Panel (c) shows a bias difference of about 0.3 m/s between the most extreme temperature classes. Although these difference are still

small, they are significant compared to the other effects described on our manuscript. Therefore we will include these new results in the final manuscript. We once again thank the referee for bringing up this effect.

Whether or not this seasonal or temperature dependent bias is due to thermal expansion of the transceiver cannot be proven here. We have no means to directly measure the focus position. We note that the wind speed distributions between summer and winter are very different, with much higher wind speeds in winter (even within the 4-16 m/s range). If the absolute wind speed error of the ZephIR would somehow increase with wind speed, this would also result in a difference in summer and winter bias, and therefore also in an (apparent) temperature trend.

The referee states that the effect of errors in the focusing height due to thermal expansion of the transceiver is well known. Unfortunately, we were not able to find publications that experimentally demonstrate the effect of thermal expansion of the transceiver, leading to larger biases, or show a seasonal or temperature dependent bias in the wind lidar wind speed data. We would be very grateful if the referee could provide some references.

AMTD

---

## Author Response (AR1)

**Response to all referee comments and list of changes**

**Referee #1 (RC1)**

(A1) "Coherent lidars also measure the shift of the frequency spectrum, albeit by a different methodology."

(A2) Probably this boils down to the question whether measuring a beat-signal is the same as measuring a shift of the frequency spectrum. However, what we had in mind is that in direct detection the frequency spectrum of the backscattered light is detected, from which the frequency shift with respect to the emitted light is determined. In coherent detection it is not the frequency spectrum of the backscattered light that is measured, but rather a beat-signal between the local oscillator (which is related to the emitted light via a frequency offset, which also can be zero).

(A3) "Direct detection wind lidars measure the frequency spectrum of the return signal", omitting "the shift of", which makes more apparent the difference between the two techniques. See **change #1.**

(B1) "It should be noted that changing the focal length also changes the depth of focus (i.e., probe length)."

(B2) A statement on the range dependent probe length was already present in the next paragraph (lines 64-65).

(B3) No change to the paper.

(C1) "It would have been interesting to find cases over the two year period where clouds existed beyond the minimum range to investigate performance of the cloud removal algorithm. "

(C2) In our comparison of the wind lidar and mast in situ data we have considered the effect of fog and low clouds (see Figure 9 of the manuscript), but we didn't specifically considered the case of only clouds below 10 m. We can select cases on basis of the visibility sensors, with the condition of fog at 2m, but no fog at all the other levels(starting at 10m). The result of the intercomparison, including this class of clouds, is included here as Fig. 1 (as provided with AC1). The "fog/low clouds" class is as defined in the paper, the new "fog below 10m" as given here. On a first glimpse the results look very good, better than the "fog/low clouds" class. However, one has to keep in mind that here wind speeds between 4 m/s and 16 m/s are considered, and that no information on the vertical thickness of the fog layer is available (rather than MOR<1000m at 2m and MOR>1000m at 10m and upwards). Thus the "fog below 10m" could include very thin and patchy cloud layers. To select events with a persistent fog layer below 10m, one needs to analyze the visibility measurements on a sub-10 minute timescale and probably lower the upper limit for the MOR. So while we agree that the point raised by the referee is interesting, a thorough analysis of it is beyond the scope of this work.

(C3) No change to the paper.

(D1) "It seems possible in the broken cloud case that the presence of clouds beyond the maximum range might be leading to more QC failures. Broken clouds might be expected to be higher and more likely to contribute to this effect."

(D2) Indeed, the way we have constructed the "low cloud" class and "broken cloud" class leaves open the possibility that in the latter case there are more clouds also above the range provided by the mast (or the wind lidar). However, the observations with the ceilometer showed that a first cloud base height above 100m has little impact on the data availability, and therefore we don't expect this property of the broken cloud class to be the cause of smaller data availability.

(D3) No change to the paper.

(E1) "Higher biases at longer range under low cloud/fog conditions could be due to asymmetric range weighting, where the signal at the more distant range of the probe volume has been attenuated relative to that at the closer, resulting in the effective measurement distance being closer to lidar than the focal range. "

(E2)This is indeed a possible reason for the observed negative bias for the fog/low clouds class. We will include this explanation in the paper.

(E3) See **change #6**.

(F1) "These are quite good results. "

(F2) Thanks for the comment.

(F3) No change to the paper.

(G1) Figure 13 (caption): "Seems like the caption is missing the word "greater" before "than 4 m/s" in line 2."

(G2) Indeed a word (either "larger" or "greater") was missing in front of the second" than 4 m/s" in this caption

(G3) See **change #9.**

**Referee #2 (RC2)**

(A1) "It is well known that thermal expansion could change the length of the transceiver telescope that would result in wind speed measurement at the wrong height. It can be interesting to check seasonal variability in the intercomparison."

(A2) The referee has raised the issue of thermal expansion of the transceiver that could lead to deviations in the position of the focus, and therefore lead to a seasonal dependent bias in the ZephIR wind speed measurements. We have analyzed the seasonal dependence. Preliminary results were provided in AC3. We find interesting results, which however, do not alter the conclusions of the paper.

Whether or not these finding link to possible temperature dependent focus height mismatch cannot be proven with our data.

The referee states that this effect of errors in the focusing height is well known. Unfortunately, we were not able to find any publications that experimentally demonstrate the effect of thermal expansion of the transceiver or textbook mentioning the effect. We have therefore decision not to explicitly mention the effect, as we cannot judge or verify its relevance. We have included now a comparison between different seasons (two summers and two winters).

(A3) See **change #7 and #8.**

(B1) "Doppler wind lidars typically cover the atmospheric boundary layer very well and thereby complement other sources of wind information, such as in-situ measurements at surface stations, weather radars, aircraft observations and satellite instruments." This is not correct. Direct detection doppler wind technique can cover a much higher range. https://journals.ametsoc.org/bams/article/86/1/73/58297https://amt.copernicus.org/articles/6/3349/2013/

(B2) We are well aware of wide variety of wind lidar instruments. The example of AEOLUS is given in our manuscript that was published online September 30, 2020, online 25 (Referee #2 unfortunately is considering an earlier version of our manuscript, which was not published online). However, we would not call the AEOLUS wind lidar "typical". Furthermore, we have stated that those instruments "cover" the boundary-layer, which does not automatically imply that that is also the limiting range.

(B3) No change to the paper.

(C1) "These wind lidars operate at a laser wavelength around 1.5 mum, and the backscatter is mainly caused by aerosols. For national meteorological services, like the Royal Netherlands Meteorological Institute (KNMI), data sets measured by these instruments can be valuable for model validation, while real-time access opens the possibility of data assimilation in operational numerical weather prediction (NWP) models and nowcasting purposes. For these applications it is of utmost importance to know the meteorological conditions in which the instruments are able to provide reliable data or not." It would be better to add a description between the different lidar techniques, e.g. direct detection, heterodyne. Also specify the used medium (aerosols, molecules)

(C2) In our manuscript that was published online September 30, 2020, we had extended the general discussion on Doppler wind lidar.

(C3) No change to the paper.

(D1) "The height range is 10-200 m above the instrument, although up to 300 m can be selected in the software." doesn't matter what you can chose in the sofware. The instrument works when aerosols are available

(D2) We agree that this instrument requires aerosols to measure wind. This is also explicitly mentioned in the introduction. However, for our location, and up to a height of 300m, enough aerosols are always present. In fact, we are not aware of limited data availability due to lack of

aerosols for short-range wind lidars (such as the ZephIR 300 or the Windcube v2).   We are not aware of publications indicating situations of too low aerosol signal for these short-range wind lidars.

(D3) No change to the paper.

(E1) "As a result, the wind lidar can resolve the wind profile better than the mast." this is pretty obvious, considering also the different costs between lidar and mast

(E2) We don't understand the point the referee wants to make here.  If the referee thinks the lidar costs are higher than the masts, then he or she underestimates the cost of mast wind measurements (including the mast infrastructure, maintenance, calibration).

(E3) No change to the paper.

(F1) "The wind lidar and mast measurements are in close agreement for most of the levels, with the exception of some part of the day where the wind speed was around25 m/s or higher, occurring mostly at 140 m and 200 m"  why those discrepancies at higher wind speeds ?

(F2)  This is indeed a very interesting observation.  Unfortunately, we have no explanation on the bias for higher wind speeds at those levels. As we wrote in Section 5.2.1, we don't think the discrepancy is due to the cup anemometers.

(F3) No change to the paper.

**List of changes (line numbering from following track change document generated by latexdiff)**

1. page 2, line 24: removed "shift of the", **in accordance to Referee 1 comment A**.
2. page 13, caption Figure 7: removed "(with and without offset)", remnant from a pre-submitted manuscript.
3. page 14, caption Figure 8: removed "without offset", remnant from a pre-submitted manuscript, changed "for" into "from".
4. page 14, line 242: include "than".
5. page 15, caption Figure 9: removed "without offset", remnant from a pre-submitted manuscript, changed "for" into "from".
6. page 15, lines 259-261: added "The presence … by the wind lidar." , **in accordance to Referee 1 comment E**.
7. page 15-16, lines 262-269: added "Finally, the extended duration … subject to further study", **in accordance to Referee 2 comment A**.
8. page 16, added Figure 10, **in accordance to Referee 2 comment A**.
9. page 20, caption Figure 14: rephrased the sentence, **in accordance to Referee 1 comment G**.
10. page 25-26, updated last access dates
11. page 25, line 394: updated reference Bosveld et al.

[revised manuscript text omitted]